# Graph Learning Indexer: A Contributor-Friendly and Metadata-Rich Platform for Graph Learning Benchmarks

**Jiaqi Ma**[*]   **Xingjian Zhang**[*]   **Hezheng Fan**[†]   **Jin Huang**[†]
**Tianyue Li**[†]   **Ting-Wei Li**[†]   **Yiwen Tu**[†]   **Chenshu Zhu**[†]   **Qiaozhu Mei**
University of Michigan

## Abstract

Establishing open and general benchmarks has been a critical driving force behind the success of modern machine learning techniques. As machine learning is being applied to broader domains and tasks, there is a need to establish richer and more diverse benchmarks to better reflect the reality of the application scenarios. Graph learning is an emerging field of machine learning that urgently needs more and better benchmarks. To accommodate the need, we introduce Graph Learning Indexer (GLI), a benchmark curation platform for graph learning. In comparison to existing graph learning benchmark libraries, GLI highlights two novel design objectives. First, GLI is designed to incentivize *dataset contributors*. In particular, we incorporate various measures to minimize the effort of contributing and maintaining a dataset, increase the usability of the contributed dataset, as well as encourage attributions to different contributors of the dataset. Second, GLI is designed to curate a knowledge base, instead of a plain collection, of benchmark datasets. We use multiple sources of meta information to augment the benchmark datasets with *rich characteristics*, so that they can be easily selected and used in downstream research or development. The source code of GLI is available at https://github.com/Graph-Learning-Benchmarks/gli.

## 1 Introduction

The practice of establishing common benchmarks in machine learning dates back to research programs of speech recognition in 1980s [1, 2] and has since become a critical cornerstone of modern machine learning research. The common benchmarking approach comes with not only a research paradigm, but also infrastructural tools (e.g., datasets, metrics, and open-source libraries) that facilitate efficient and effective iterations of machine learning research. In the past, the community has been focusing on a handful of benchmarks in each major domain of machine learning applications[3], usually developed by few institutes or research groups [6]. However, as machine learning is becoming a general-purpose technology, there are *new demands* from modern machine learning research that are not entirely met by the current common practice of benchmarking:

1. *Breadth of Applications.* Machine learning is applied to increasingly broader domains. The emerging field of graph learning is an example where there exist a variety of machine learning tasks. Representative new benchmarks are needed for such emerging domains and tasks, and the development of good benchmarks often requires interdisciplinary knowledge and collaboration.

2. *Trustworthiness.* Each individual benchmark dataset is likely to be biased due to certain ad-hoc design choices in the data collection process. Driving the development of machine learning technologies with a couple of fixed benchmark datasets poses a risk of having trustworthy

---

[*]Equal contribution. Contact: jiaqima@{umich.edu, illinois.edu}; jimmyzxj@umich.edu.
[†]Alphabetical order.
[3]For example, ImageNet [3] in Computer Vision, SuperGLUE [4] in Natural Language Processing, and Open Graph Benchmark [5] in Graph Learning.

J. Ma and X. Zhang et al., Graph Learning Indexer: A Contributor-Friendly and Metadata-Rich Platform for Graph Learning Benchmarks. *Proceedings of the First Learning on Graphs Conference (LoG 2022)*, PMLR 180, Virtual Event, December 9–12, 2022.

issues ignored by the limited number of benchmarks. Leveraging a set of *diverse* datasets for benchmarking can mitigate this risk by exposing more potential trustworthy concerns at the early benchmarking stage.

3. *Task Generalizability.* Increasingly towards general-purpose artificial intelligence, there is a strong trend in developing machine learning models that can perform well on a wide range of downstream tasks [7]. In conjunction with this interest, there have been efforts constructing benchmarks with *many tasks*, such as SuperGLUE [4], GEM [8], and BIG-Bench [9], where BIG-Bench consists of 204 tasks by more than 400 authors across 132 institutes.

These new demands, especially for emerging fields such as graph learning, require the development of massive and diverse benchmark datasets in order to better reflect the reality of machine learning applications. This requirement poses technical challenges in both the creation and the curation of benchmarks, which calls for novel infrastructural tools to facilitate the benchmark research.

In this paper, we introduce Graph Learning Indexer (GLI), a graph learning benchmark curation platform, to mitigate the aforementioned challenges. In particular, GLI highlights two novel design objectives that respectively mitigate the challenges in benchmark creation and curation.

First, GLI aims to leverage contributions from the broad graph learning community to establish a wide range of benchmarks. As a result, GLI is designed to be *contributor-centric*, where we treat benchmark contributors as our core users when designing the platform. Specifically, we incorporate various designs, such as file-based data API, automated test, and template files, to minimize the effort of contribution and maintenance by the benchmark contributors. We have also considered measures to incentivize research efforts in benchmark contributions in general. For example, in order to encourage better attributions to the benchmark contributors, GLI includes the chain of prior versions of each benchmark dataset in the bibliographic section of the dataset README file.

Second, with the increasing quantity and diversity of benchmark datasets, GLI aims to build a knowledge base where every dataset is augmented with rich metadata, instead of a plain collection of datasets. GLI includes a *Benchmark Indexing System*[4] with various sources of meta information about the benchmark datasets collected by GLI. Such meta information can be later used for better curation and retrieval of the benchmarks.

The rest of this paper is organized as follows. We introduce the contributor-centric design and the benchmark indexing system respectively in Section 2 and Section 3. Section 4 reviews related prior work on benchmark collections and graph learning libraries. We also include a sketch of future plan for GLI in Section 5. Finally, in Section 6, we conclude this paper with some open questions.

## 2 Contributor-Centric Design

A central goal of GLI is to incentivize the community to put more effort into contributing high-quality benchmark datasets. To achieve this goal, we treat dataset contributors as the core users of GLI and come up with three contributor-centric design objectives. First, GLI aims to provide a smooth user experience for contributors by minimizing the effort in the submission and maintenance of the datasets. Second, GLI aims to increase the impact of the hosted datasets by improving their usability. Third, GLI aims to encourage better attributions to the dataset contributors through tangible measures.

### 2.1 User Experience and Quality Assurance

A key challenge in the design of GLI is to minimize the effort by the dataset contributors while assuring a high quality of the contributed datasets. Our solution to this challenge is to first design a standard data management API that is both stable and extensible for graph learning datasets; and then design a GitHub-based contribution workflow with concise instructions and rich feedback for dataset contributors to convert the benchmark datasets into the standard API.

### 2.1.1 Data Management API

The GLI Data Management API (Figure 1) has two key design features: the API is file-based; there is an explicit separation of data and task.

---

[4]Thus "Indexer" in the name of GLI.

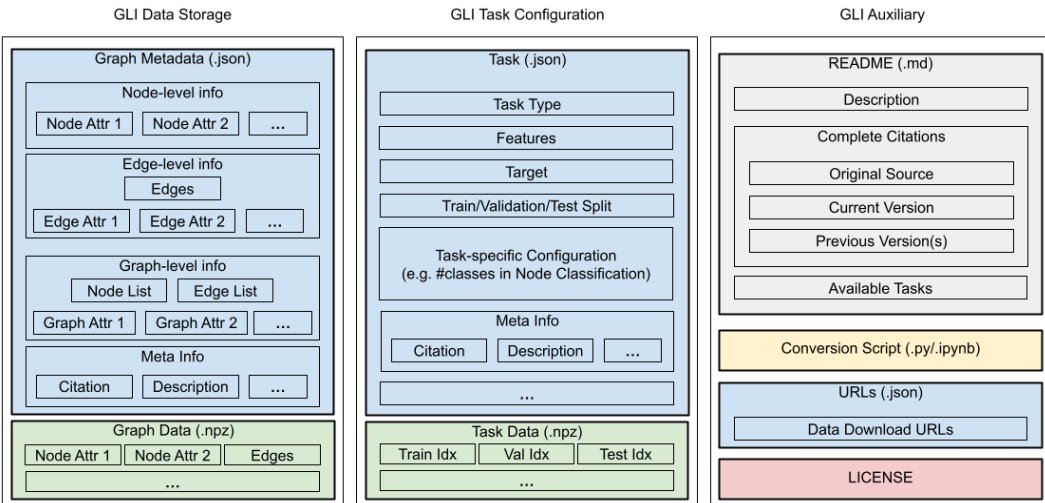

**Figure 1:** The file-based *GLI Data Management API* with explicit separation of data and task. The *GLI Data Storage* part contains all the necessary information to construct the graph data, including three levels: node, edge, and graph information. Each level may have multiple features or labels as its attributes. The *GLI Task Configuration* part contains the necessary information to perform a predefined task. Both parts further compress big chunks of data (such as the attributes or edge list) into NumPy standard binary format, with indexes to these data stored in JSON files. The NumPy data files are hosted in an external storage system, while all other files are hosted in the GitHub repo of GLI. In addition, the *GLI Auxiliary* part contains a README document, a conversion script that converts the raw data into GLI file format, a LICENSE file, and a `urls.json` providing the URLs to the NumPy data in the external storage system.

**File-based storage API.** The data API for almost all existing graph learning libraries (such as DGL [10] and PyG [11]) are code-based, which means that each dataset is associated with an ad hoc class that is dedicated to representing this dataset. For example, DGL [10] defines a `CoraGraphDataset` class for the node classification task on the Cora dataset [12–15]. This code-based API couples the datasets with the codebase and increases the difficulty of maintenance. In particular, changes to the graph learning library codebase may break the ad hoc dataset classes so additional maintenance effort is required for each dataset.

To avoid such unnecessary maintenance burden for dataset contributors, GLI adopts a *file-based* data storage API that is more stable compared to code-based APIs. While there has been file-based graph storage API, such as GraphML [16], they are not dedicated to graph learning datasets and lack essential features such as storing the data splits. We therefore designed a novel file-based storage API for graph learning datasets.

**Explicit separation of data and task.** We recognize that there is a clear distinction between the information of the content in a dataset, i.e., the *data*, and the information about how to use the data to train and evaluate the models, i.e., the *task*. For example, in graph learning benchmarks, there could often be multiple tasks (e.g., node classification and link prediction) defined on the same dataset, or there could be multiple settings for the same task (e.g., random split or fixed split). From the perspective of dataset contribution and curation, it is cumbersome to create a new dataset version for each new task on top of the same data. Therefore, we propose to store the *data information* and the *task information* separately in our API. And we design a task-specific API for each type of tasks.

This explicit separation of data and task turns out to offer a number of benefits. First, it makes the API more extensible, as the introduction of a new type of task will not affect the API for the data. Second, this separation makes automated tests more modularized (see Section 2.1.2). Third, it allows the implementation of general data loading schemes (see Section 2.2). Finally, it leads to a bottom-up approach to growing the taxonomy of graph learning tasks (see Section 3.1).

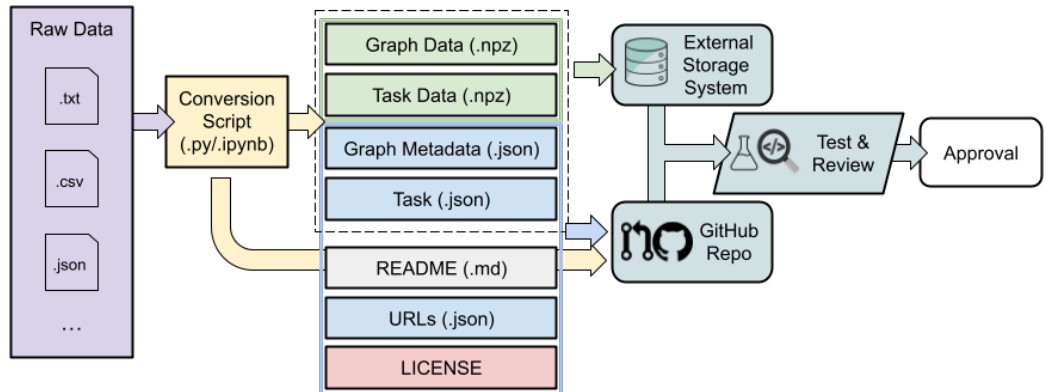

**Figure 2:** GLI Contribution Workflow. A contributor will first use the conversion script to convert the raw data into the GLI format. Then the contributor will fill in the templates of `README.md` and `urls.json`. The JSON files and auxiliary files (blue arrow), and the conversion script (yellow arrow) will be uploaded to GitHub as a pull request and the NumPy data files (green arrow) will be uploaded to the external storage system. GLI will perform automated tests on the submitted datasets and the GLI development team will further review the pull request before approval.

**Overview of the API.** Figure 1 shows the architecture of the file-based API with explicit separation of data and task[5].

The information of the graph data is divided into three levels: node, edge, and graph level. Each level can be assigned multiple attributes as features or labels and can be further divided into multiple sub-levels to represent heterogeneous graphs. The attributes support both dense and sparse tensors to allow efficient storage and fast loading. The GLI data format has a strong representative power to accommodate most graph-structured data.

For the task, we have predefined a number of graph learning task types, such as `NodeClassification`, `LinkPrediction`, `GraphClassification`, etc. The information in the task configuration can be divided into two kinds: general configuration and task-specific configuration. General configurations are commonly required by all tasks, including features that are allowed to use during prediction, train/validation/test split, etc. On the contrary, the contents of task-specific configurations depend on task types. For example, both `NodeClassification` and `GraphClassification` require to specify the number of possible classes (num_classes), and `LinkPrediction` provides an optional configuration on negative samples during validation and test (val_neg and test_neg).

Overall, the file-based design improves the stability of the API while the separation of data and task makes the API more extensible, both in turn improving the user experience for dataset contributors.

### 2.1.2 Contribution Workflow

In companion with the data management API, we designed a GitHub-based contribution workflow (Figure 2) to ease the dataset contribution process.

**Template files.** To begin with, GLI provides a list of well-commented template files[6] for all the required files in our API. The contributor only needs to fill in all the blanks to convert a dataset into the GLI format.

**Dataset submission and review.** After finishing converting the dataset, the contributor will submit the required files as a pull request to the GitHub repository of GLI. The large NumPy binary files

---

[5]A detailed document for the API is available at `https://github.com/Graph-Learning-Benchmarks/gli/blob/main/FORMAT.md`.

[6]See `https://github.com/Graph-Learning-Benchmarks/gli/tree/main/examples/template`.

will be uploaded to an external storage system[7]. The GLI development team or other researchers can provide detailed and interactive feedback in the pull request.

**Automated tests.** In addition to the manual peer review, the pull request will also trigger automated tests with detailed error feedback to help the contributors debug their implementation. The tests include the standard `pycodestyle`, `pydocstyle`, and `pylint` for syntax and style checks. We have also implemented a wide range of in-depth tests with `pytest` to check the correctness of dataset format and to expose potential errors during runtime by sanity check with short model training. Contributors can also use several well-documented utility functions to test the correctness of their data format locally.

## 2.2 Dataset Usability

```
import gli

cora_node_dataset = gli.get_gli_dataset(dataset="cora",
                                        task="NodeClassification")
```

**Demo 1:** Example usage of the general data loading scheme. `cora_node_dataset` is an instance of `dgl.data.DGLDataset`, thus it can be fit into DGL dataloader seamlessly.

To increase the impact of the datasets hosted on GLI, we implemented a general task-centric data loading scheme that can be seamlessly integrated into major graph learning libraries for downstream experiments. At the time of writing this paper, we have implemented data loading for DGL [10]. We also strive to accommodate other major libraries in the future. Demo 1 demonstrates an example of the general data loading scheme. Once a contributed dataset (and the task defined on it) is merged into the GLI repository, the dataset can be retrieved by calling `gli.get_gli_dataset` with the dataset name and task type as arguments.

Under the hood, as shown in Demo 2, `gli.get_gli_dataset` calls `gli.get_gli_graph` and `gli.get_gli_task` to respectively load the GLI Data Storage and GLI Task Configuration shown in Figure 1. Thanks to the explicit separation of data and task, we only need to maintain a general graph loading function and a set of task loading functions with each function dedicated to a task type, which is much less effort than maintaining a separate dataset class for each task and dataset combination.

```
graph_cora = gli.get_gli_graph(dataset="cora")
task_node = gli.get_gli_task(dataset="cora",
                             task="NodeClassification")
cora_node_dataset = gli.combine_graph_and_task(graph_cora, task_node)
```

**Demo 2:** The inner-workings of `gli.get_gli_dataset`.

## 2.3 Attributions to Contributors

An important aspect to incentivize the dataset contributors is to ensure that they get the proper credits. For this purpose, we have made a couple of designs to help the community cite properly. There is citation information in the README file of each dataset listing the BibTex of the work relevant to the dataset. Specifically, the citation information is split into dataset and tasks, as there could often be multiple tasks defined on top of a graph dataset, and the definition of tasks could come from work that is different from the one contributing to the dataset. Moreover, the citation information for the dataset is further split into three parts:

- *Original Source*: The first work that created the dataset.

- *Current Version*: The work that is directly responsible for the dataset stored in GLI.

- *Previous Versions*: Any intermediate versions between Original Source and Current Version. There can be multiple citations in Previous Versions.

---

[7]Currently we use Dropbox accounts owned by the GLI development team as the storage system.

The paper popularizing a benchmark dataset is often not the paper originally contributing the dataset. And it is not uncommon that the former gets most of the citations while the latter gets few[8]. This phenomenon is possibly due to two factors. First, tracking the chain of contributions to a dataset through a literature search is a tedious job. Second, researchers tend to get information about a dataset from the methodology papers that cite the dataset rather than the original paper creating the dataset. So the mistakes in citations accumulate.

By providing succinct bibliographic information relevant to the dataset in the README file, we hope to help the community better recognize the contributions of all contributors, with a particular emphasis on crediting the original source.

## 3 Benchmark Indexing System

With the growing quantity and diversity of benchmark datasets, it is important for the benchmark curation platform to help users efficiently navigate through the large collection of datasets. For this purpose, GLI is designed to serve as an "indexer" that builds a database consisting of various meta information of the benchmark datasets. And we name the database as *Benchmark Indexing System*. To some extent, this is in a similar vein as the idea of *Datasheets for Datasets* [17]. Datasheets for Datasets focus more on the characteristics of each individual dataset while our design of the database also cares about the synergy among different datasets.

Ultimately, we hope to use this database to help users 1) retrieve the right benchmarks that match the context of the applications of their interest; 2) identify potential biases and trustworthy issues existing in the datasets; or 3) motivate the development of new methodology based on the characteristics of tasks and datasets.

At the current stage, however, we focus on coming up with different sources of meta information to be included in the database. The current implementation consists of three types of meta information, which are detailed in the following subsections.

### 3.1 Task Types

The task types come as meta information naturally from the implementation of data management API in GLI. Graph data are ubiquitous but also diverse and so are the graph learning tasks defined on top of graph data. Different graph learning tasks may have distinct natures and thus require very different methodologies. Therefore the task type is an important source of meta information for each benchmark dataset.

In GLI, the definition of task types is driven by the contributed benchmarks. When a contributor is contributing a new benchmark, they will first check if their benchmark belongs to one of the existing task types in GLI. If none of the existing task types can accommodate the new benchmark, the contributor can initiate the definition of a new task type. The GLI development team and the contributors will implement the support for the new task type, including dataset class, documentation, and automated tests.

This bottom-up approach of developing task types not only makes GLI highly extensible to new benchmark datasets, but also gradually grows a taxonomy of graph learning tasks as more benchmarks are being collected. A list of currently supported task types is given in Appendix A.

### 3.2 Graph Data Properties

Another type of meta information included in GLI is various graph data properties, such as average degree or average clustering coefficient. In classical network science literature [18, 19], the graph data properties have been shown to be informative about the characteristics of the graph data. In a recent study, Palowitch et al. [20] empirically demonstrated that there are clear patterns in the graph neural network performance associated with certain graph data properties of the benchmark datasets.

GLI integrates a function that can calculate a list of graph data properties for each contributed dataset. These graph data properties can be categorized into 6 groups.

---

[8]This happens even for very popular datasets. See Appendix B.1 for a case study.

- *Basic*: Is Directed, Number of Nodes, Number of Edges, Edge Density, Average Degree, Edge Reciprocity, Degree Assortativity;

- *Distance*: Diameter, Pseudo Diameter, Average Shortest Path Length, Global Efficiency;

- *Connectivity*: Relative Size of Largest Connected Component (LCC), Relative Size of Largest Strongly Connected Component (LSCC), Average Node Connectivity;

- *Clustering*: Average Clustering Coefficient, Transitivity, Degeneracy;

- *Distribution*: Power Law Exponent, Pareto Exponent, Gini Coefficient of Degree, Gini Coefficient of Coreness;

- *Attribute*: Edge Homogeneity, Feature Homogeneity, Homophily Measure, Attribute Assortativity.

The formal definitions of these graph data properties can be found in Appendix C.

## 3.3 Model Performance

The third type of meta information included in GLI is the performance of various popular models on the datasets. It is common to use a model's performance on different experiment settings and datasets to understand the model characteristics. Recently, it is shown that one can also use the performance of different models to characterize the datasets and obtain meaningful clusters of the datasets [21].

In GLI, we provide a benchmark suite that can benchmark a few popular machine learning models on the contributed benchmarks. The benchmark suite implements a separate set of training and hyperparameter tuning functions for each task type. Thanks to the general data loading scheme (as introduced in Section 2.2), the benchmark code can be easily extended to new datasets with the same task type. We currently have supported `NodeClassification` and `GraphClassification` in the benchmark suite.

Below, we provide an example to showcase how the model performance could provide useful information to characterize the datasets. Using the benchmark suite in GLI, we provide the performance of several popular models on a set of node classification datasets in Table 1. This experiment is a rough replication of Lim et al. [22], with an extension to more datasets enabled by GLI. The detailed experiment setup (and citations to models and datasets) can be found in Appendix D.

Readers who are familiar with the recent graph learning literature may find that, not surprisingly, the best and second best performing models on each dataset are a good indicator of how "homophilous" [23] the dataset is. The early graph neural network models, GCN, GAT, and GraphSAEG, have better performance on more homophilous datasets, such as cora, citeseer, and pubmed. LINKX performs better on most of the remaining non-homophilous datasets. A few datasets, texas, cornell, and wisconsin, lead to notoriously unstable model performance, as shown by the large standard deviations for most models. It also seems that the graph structure does not help much for the task, as the models (MLP, CatBoost, and LightGBM) that do not utilize the graph structure perform the best on these datasets.

In general, the GLI API makes it easier to implement the benchmark suite for a wide range of models and datasets in well-controlled experiment setups, which enables the use of model performance as a way to characterize the datasets.

## 4 Related Work

In this section, we review prior work on graph learning benchmarks, graph learning libraries, and other relevant efforts on machine learning benchmark infrastructures.

### 4.1 Graph Learning Benchmarks and Graph Learning Libraries

Recently, there have been many infrastructural efforts on developing benchmark collections for graph learning [5, 24–27]. Among which the most widely-used ones at present are perhaps Open Graph Benchmark [5] and Benchmarking Graph Neural Networks [25]. GLI differs from the prior work in two key aspects.

**Table 1:** Benchmark experiment results for node classification datasets. Test accuracy is reported for most datasets, while test ROC AUC is reported for binary classification datasets (genius, twitch-gamers, penn94, pokec). Standard deviations are over 5 runs. The best result on each dataset is bolded, and the second-best result is underlined.

| | GCN | GAT | GraphSAGE | MoNet | MLP | CatBoost | LightGBM | LINKX | MixHop |
|---|---|---|---|---|---|---|---|---|---|
| cora | 81.03±0.82 | **83.0±0.62** | 81.46±0.74 | 76.44±1.85 | 59.1±2.3 | 59.38±1.25 | 36.40±0.00 | 59.36±2.41 | 79.64±1.55 |
| citeseer | 72.28±0.56 | 69.9±1.54 | **73.38±0.82** | 64.4±0.62 | 54.62±6.26 | 59.18±0.58 | 39.34±0.77 | 42.5±7.88 | 69.64±1.2 |
| pubmed | **79.44±0.43** | 79.04±0.76 | 78.4±0.35 | 76.18±0.84 | 73.7±0.5 | 69.96±1.15 | 54.86±0.33 | 56.49±7.92 | 76.61±1.35 |
| texas | 61.08±3.07 | 67.02±1.21 | 66.48±1.48 | 55.13±7.04 | 78.92±2.25 | 77.84±1.21 | **83.78±0.00** | 76.57±4.87 | 77.84±1.7 |
| cornell | 52.97±4.09 | 48.64±1.9 | 47.02±3.08 | 51.89±2.25 | 68.64±7.78 | 69.19±2.42 | **77.30±1.48** | 65.46±5.85 | 66.48±5.43 |
| wisconsin | 56.46±3.5 | 54.89±1.96 | 52.54±1.63 | 36.86±3.22 | 78.82±4.24 | 81.18±2.24 | **81.96±0.88** | 78.62±1.94 | 76.9±5.61 |
| actor | 29.36±0.73 | 30.15±0.56 | 29.26±0.5 | 26.35±1.01 | **37.11±0.54** | 34.57±1.44 | 32.12±0.24 | 33.56±1.84 | 34.77±0.94 |
| squirrel | 32.4±1.18 | 29.14±1.55 | 31.64±1.93 | 27.14±2.34 | 34.87±0.47 | 34.37±0.37 | 33.89±0.69 | **62.43±1.23** | 33.37±1.45 |
| chameleon | 45.92±2.61 | 46.18±0.93 | 48.72±0.47 | 32.54±1.24 | 49.16±0.66 | 41.89±2.54 | 30.92±1.24 | **67.08±1.69** | 48.72±1.39 |
| arxiv-year | 49.6±0.16 | 34.91±0.56 | 43.39±0.74 | 40.19±0.48 | 36.49±0.19 | 35.76±0.60 | 36.17±0.29 | **52.73±0.34** | 40.63±0.12 |
| snap-patents | **55.46±0.11** | 36.34±0.6 | 43.33±0.27 | 43.48±0.73 | 31.32±0.04 | 30.96±0.55 | 31.48±0.06 | 53.43±0.32 | 43.27±0.03 |
| penn94 | 88.79±0.6 | 66.29±12.21 | 85.0±0.53 | 73.92±3.71 | 83.92±0.32 | 73.21±2.20 | 73.62±0.05 | **93.47±0.27** | 91.62±0.11 |
| pokec | 71.17±10.76 | 53.03±0.4 | 63.02±5.68 | 53.65±2.17 | 64.69±4.92 | 62.55±0.38 | 62.77±0.03 | **90.54±0.12** | 86.84±0.2 |
| genius | 84.15±1.71 | 49.86±28.68 | 80.31±0.23 | 63.23±2.39 | 84.42±0.2 | 82.48±0.00 | 82.48±0.00 | **90.88±0.1** | 90.04±0.12 |
| twitch-gamers | 62.4±0.22 | 59.57±0.88 | 61.68±0.3 | 58.02±1.26 | 59.66±0.09 | 61.57±0.05 | 61.62±0.02 | **66.21±0.3** | 64.22±0.08 |

1. GLI is specifically optimized to better serve the dataset contributors. Most existing graph learning benchmarks are designed with the "dataset consumers", instead of contributors, as the core users. To our best knowledge, dedicated designs to optimize the contribution workflow of graph learning datasets were essentially non-exist prior to this work. For example, the contribution workflow for Open Graph Benchmark is to pack the dataset in a fixed format and email it to the maintenance team[9]. In comparison, our GitHub-based contribution workflow is more interactive and potentially more scalable.

2. GLI maintains a bottom-up dynamic task taxonomy while most of the existing benchmark collections have a top-down static taxonomy of graph learning tasks. The static taxonomy of graph learning tasks may limit the type of dataset and tasks that could be contributed to the benchmark collections.

There are also a few workshops and conference tracks dedicated to research on benchmarks and datasets, such as the Workshop on Graph Learning Benchmarks[10] and the NeurIPS Datasets and Benchmarks Track[11]. These venues are friendly to the publications of benchmark contributions and have successfully solicited a number of new graph learning benchmark datasets. The development of GLI shares the same motivation as these endeavors towards incentivizing more contributions on benchmarks. And GLI could be used as an infrastructural tool for these publication venues to better evaluate and curate the collected benchmarks.

## 4.2 Graph Learning Libraries

In addition, there are a few general-purpose graph learning libraries, such as PyG [11], DGL [10], and TF-GNN [28], that are relevant to this work. While the primary focus of these libraries is not on benchmark datasets, they also provide graph data API at the dataloader level. The file-based API design in GLI is more contributor-friendly because 1) it is easier to convert the data to files than to implement a dataset class; 2) the file-based API does not rely on any software dependency and is less likely to break; 3) the GLI developers will take care of the maintenance of the data loading code.

## 4.3 Other Relevant Benchmark Infrastructures

Outside the area of graph learning, there are various machine learning benchmark infrastructures that are remotely relevant to this work.

One relevant machine learning benchmark infrastructure is Papers With Code[12], which has a database of datasets in different domains of machine learning. Each dataset in this database is associated with types of machine learning tasks and a massive record of machine learning model performances,

---

[9]https://ogb.stanford.edu/docs/dataset_overview/.

[10]https://graph-learning-benchmarks.github.io/.

[11]https://neurips.cc/Conferences/2021/CallForDatasetsBenchmarks.

[12]https://paperswithcode.com/.

similar to our design in Section 3. However, the performances are directly taken from papers or self-reported, and the experiment setups and data versions may not be well controlled.

More generally, there are a number of dataset search engines, such as Google Dataset Search[13], Microsoft Research Open Data[14], and DataMed[15]. These search engines index a large amount of datasets in various domains but do not contain detailed domain-specific characteristics, such as the graph metrics as described in Section 3.2. These datasets are also usually not machine-learning ready, i.e., there is no data loading code that transforms these datasets into machine learning data loaders.

Finally, there are many community organizations and efforts for creating benchmarks, such as TREC[16] for Information Retrieval, and CLEF[17] for Natural Language Processing. The infrastructural tools developed in GLI can also potentially be adapted to support the community efforts outside the graph learning domain.

## 5 Future Plan

In the near future, there are a few directions that the GLI development team will focus on.

**User experience.** There is still room to further simplify the dataset contribution workflow, which will be one of the major focuses in our future development plan. As examples, we have planned to work on the following concrete improvements.

- *Helper functions for dataset conversion.* We plan to implement a few helper functions that can automatically convert commonly seen raw data formats into the GLI format.

- *Automatic generation of README documents.* We would like to implement a function that can automatically generate the README document for a dataset based on dataset characteristics and a few structured survey questions for the contributors.

- *Simplified submission interface.* While the Pull Request functionality of GitHub offers many advantages for dataset review (such as providing tests and reviews, and preserving review and commit history), the additional technical complexity brought by this process may be a concern. In the future, we may want to explore methods to automatically construct a Pull Request based on a simpler dataset submission interface.

**Automatic benchmarking popular models.** We plan to implement a service that can automatically benchmark popular models on newly contributed datasets such that the model performance can be directly leveraged into the meta information of the datasets.

**Citation tracking.** We plan to track the citations to each dataset hosted on GLI. In this way, we can send an alert to the authors citing a dataset when critical issues/bugs are identified for the dataset.

**Dataset exploration.** We plan to implement an interface to explore and retrieve the datasets hosted on GLI, based on the database of the datasets described in Section 3.

**Dataset license.** A surprisingly large number of commonly used datasets lack an explicit license associated with them. Moreover, getting the right license for many existing datasets is a complicated task for a few reasons. First, many commonly used datasets, especially those created in the early years, do not have a license. Second, while some datasets have a license, they are repurposed from an early version that does not have a license. It is unclear if such licenses are still valid. Finally, many datasets are released within a code repo. It is unclear if the license of the code repo could be viewed as the license to the datasets. As an important future step, we plan to take various measures to mitigate the license problem for datasets hosted on GLI. In particular, we will implement automated tools to enforce the license coverage for newly contributed datasets. We will also provide guidance on license choices for dataset contributors.

---

[13]https://datasetsearch.research.google.com/.
[14]https://msropendata.com/.
[15]https://datamed.org/.
[16]https://trec.nist.gov/.
[17]https://www.clef-initiative.eu/.

# 6    Conclusion

In this paper, we have introduced Graph Learning Indexer (GLI), an infrastructural tool for benchmark research in graph learning. GLI is designed to solicit and curate massive benchmark datasets contributed by the community. With the contributor-centric design, GLI can better assist the community contribution to the development of benchmark datasets. With the rich metadata of datasets annotated, GLI can help us improve our understanding of the taxonomy of graph learning tasks, as well as better navigate through the massive datasets. The development of GLI will also be a long-term endeavor.

Finally, to conclude this paper, we raise a few interesting open problems motivated by our effort in building infrastructural tools for benchmark research in this work.

## 6.1    Open Problems

**A "TEX" system for data publication.**    The TEX typesetting system automates the process of typography and allows article authors to focus on the content without worrying about the layout of the article. The TEX files can also be easily reused for different templates. As publishing code and data alongside the papers has become a common practice in machine learning research, it is valuable to develop a "TEX" system (analogously) for data publication. Ideally, there could be a set of common syntax for storing the data and metadata of a published dataset. Datasets stored in this syntax can be easily reused/loaded into different machine learning pipelines. Automatic analysis/diagnosis can be performed on the dataset to extract more standard meta information about the dataset. GLI can be viewed as an early attempt towards such a "TEX" system for data publication.

**Quality assurance of benchmark datasets.**    While there has been much effort in evaluating the quality of machine learning models, in terms of both prediction accuracy and various trustworthiness metrics, evaluating the quality of datasets is relatively under-explored. Recently, there have been efforts dedicated to understanding and improving the data quality under the name of *Data-Centric AI*[18]. In general, quality assurance is still a critical open problem for benchmark dataset curation. GLI has implemented low-level automated tests ensuring the correctness of data storage formats. It would be helpful to further introduce high-level quality metrics such as the signal-to-noise ratio of the data labels. Crediting the contributors through a chain of prior versions can also incentivize the community to keep improving the quality of a dataset.

**Infrastructure for efficient reproducibility.**    Reproducibility has been an important issue in machine learning and data science research in general. In addition to scholarship factors, there are two practical challenges hindering reproducibility. First, the complex experimental configurations in machine learning and data science research require tedious efforts to exactly reproduce a result and/or have a fair comparison between methods. Second, reproducing some of the results has considerable computational costs. There is a huge open opportunity for developing infrastructural tools to mitigate these practical challenges in reproducibility. In particular, designing unified data and task configuration APIs in combination with cloud-based computing infrastructures (such as CodaLab[19] or LiveDataLab [29]) may be a promising direction.

**Co-evolution of models and benchmarks.**    A fundamental problem with the existing benchmark-driven machine learning research paradigm arises from the well-known Goodhart's law [30]: "*when a measure becomes a target, it ceases to be a good measure*" [31]. A fixed set of benchmarks can be quickly overfitted and become no longer meaningful for evaluating new models [32]. It is therefore desirable for the benchmarks to evolve at a pace that matches the development of new models. Easing the process of dataset contributions is an important first step toward this goal.

## Author Contributions

JM initiated the project. JM, QM, and XZ designed the scope of the project. JM and XZ came up with the concrete design of the GLI API, contribution workflow, and benchmark indexing system. XZ designed and implemented most of the core dataloading functionality of GLI. HF, TL, YT, and CZ converted most of the datasets into the GLI format. HF, JH, and CZ designed and implemented

---

[18]https://datacentricai.org/.
[19]https://codalab.org/.

the majority of automated tests. TWL was responsible for implementing and obtaining the graph data properties for all datasets. JH designed and implemented most of the benchmark code and ran half of the benchmark experiments. TL also significantly contributed to the benchmark code and the benchmark experiments. XZ was responsible for keeping track of the to-do and bug issues. XZ, JM, and JH took care of most of the code review. XZ generated the project website. JM and XZ wrote the documentation. Finally, JM, XZ, and QM wrote this paper. Most authors also lightly cross-contributed to other parts.

## Acknowledgements

The authors would like to thank Danai Koutra, Anton Tsitsulin, ChengXiang Zhai, and Jiong Zhu for helpful discussions, all the participants in the GLB 2021 and GLB 2022 workshops for motivating this project, and anonymous reviewers at LOG 2022 for constructive suggestions. This work was in part supported by the National Science Foundation under grant number 1633370.

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

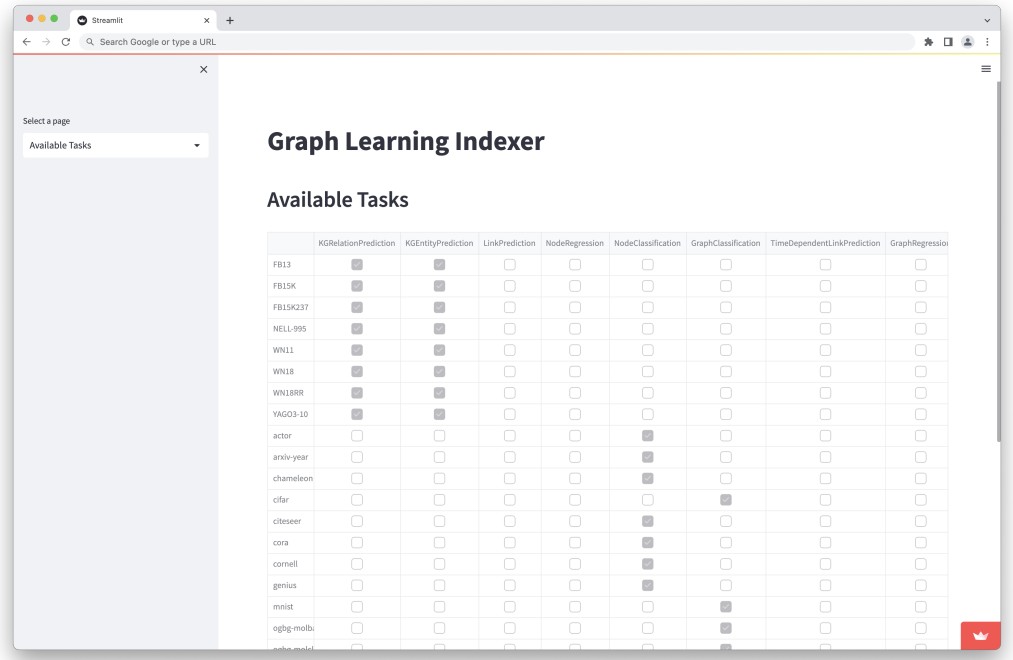

**Figure 3:** Available tasks table on GLI web page. The rows are datasets and the columns are pre-defined task types.

## A  List of Task Types

Currently, GLI supports the following task types[20]:

1. `NodeClassification`: Node classification task. This task aims to predict categorical node properties based on other nodes and its features in a graph.

2. `NodeRegression`: Node regression task. This task aims to predict continuous node properties based on other nodes and its features in a graph.

3. `GraphClassification`: Graph regression task. This task aims to predict categorical graph properties based on known graph's features.

4. `GraphRegression`: Graph classification task. This task aims to predict continuous graph properties based on known graph's features.

5. `LinkPrediction`: Link prediction task. This task aims to predict the existence of a link between two nodes in a graph.

6. `TimeDependentLinkPrediction`: Link prediction task, split by time. This task is the special case of `LinkPrediction`. Its train-validation-test split depends on the creation time of links.

7. `KGEntityPrediction`: Knowledge graph entity prediction task. This task aims to predict the tail or head node for a triplet in the graph.

8. `KGRelationPrediction`: Knowledge graph relation prediction task. This task aims to predict the relation type for a triplet in the graph.

The supported tasks for each dataset is shown in a table on our web page, as can be seen in Figure 3. This page will be updated periodically to include new task configurations contributed to GLI.

---

[20]https://github.com/Graph-Learning-Benchmarks/gli/blob/main/FORMAT.md#gli-task-format.

## B  Reference of Datasets

Table 2 summarizes the original source, current version and previous versions of the datasets that we have incorporated.

**Table 2:** Reference of datasets.

| Dataset | Original | Cur | Prev | Dataset | Original | Cur | Prev |
|---|---|---|---|---|---|---|---|
| actor | [33] | [34] | / | ogbg-molpcba | [35] | [5] | [36] |
| arxiv-year | [37, 38] | [22] | [5] | ogbl-collab | [37] | [5] | / |
| chameleon | [39] | [34] | / | ogbn-arxiv | [37] | [5] | / |
| cifar | [40] | [25] | / | ogbn-mag | [37] | [5] | / |
| citeseer | [13, 41] | [15] | [14] | ogbn-products | [42] | [5] | [43] |
| cora | [12, 13] | [15] | [14] | ogbn-proteins | [44] | [5] | [45] |
| cornell | [46] | [34] | [13] | penn94 | [47] | [22] | / |
| FB13 | [48] | [49] | [50] | pokec | [51, 52] | [22] | / |
| FB15K | [48] | [49] | [53] | pubmed | [54] | [15] | / |
| FB15K237 | [48] | [49] | [53] | snap-patents | [51] | [22] | [55] |
| genius | [56] | [22] | / | squirrel | [39] | [34] | / |
| mnist | [57] | [25] | [58] | texas | [46] | [34] | [13] |
| NELL-995 | [59] | [60] | [49] | twitch-gamers | [61] | [22] | / |
| ogbg-molbace | [35] | [5] | [36] | wiki | [22] | [22] | / |
| ogbg-molclintox | [35] | [5] | [36] | wiscousin | [46] | [34] | [13] |
| ogbg-molfreesolv | [35] | [5] | [36] | WN11 | [62] | [49] | [63] |
| ogbg-molhiv | [35] | [5] | [36] | WN18 | [62] | [49] | [63] |
| ogbg-molmuv | [35] | [5] | [36] | WN18RR | [62] | [49] | [63] |
| ogbg-molsider | [35] | [5] | [36] | YAGO3-10 | [64] | [49] | [65] |

### B.1  A Case Study on the Citations of Cora, CiteSeer, and PubMed

Properly crediting the dataset contributors, unlike many may imagine, could be surprisingly non-trivial and require much more than good intentions of authors. In this section, we illustrate the challenges with a case study on the citations of the three popular datasets, Cora, CiteSeer, and PubMed.

First, many authors tend to cite only *some* relevant literature, instead of *all* relevant literature. The Planetoid [15] version of Cora, CiteSeer, and PubMed is the one that got popularized and mostly used. There is a significant number of papers that only cite Yang et al. [15].

Second, mistakes in citations will be inherited and cascaded. It is common in the graph learning literature that Cora, CiteSeer, and PubMed are attributed to Sen et al. [14]. In fact, Sen et al. [14] only introduced Cora and CiteSeer but not PubMed. PubMed should be attributed to Namata et al. [54] instead.

Third, it could be tricky to define the "original source" of a dataset. Taking the history of the Cora dataset as an example, Cora can be at least traced back to McCallum et al. [12], who developed an Internet portal, named "Cora", that organizes a collection of computer science research papers under a topic hierarchy, with citation links and bibliographic information available. Lu and Getoor [13] constructed a paper classification dataset based on the collection by McCallum et al. [12]. While the dataset by Lu and Getoor [13] is extracted from the data collection by McCallum et al. [12], to our best knowledge, Lu and Getoor [13] is the first work that established the task of predicting paper categories using the paper content information and the citation links among papers. Sen et al. [14] and Yang et al. [15] consecutively made minor changes to the dataset by Lu and Getoor [13], which leads to the Cora dataset that is widely used in the graph learning community nowadays. It is tricky to decide whether McCallum et al. [12] or Lu and Getoor [13] should be considered as the original source of the Cora dataset – the former contributed the raw data collection, while the latter made a significant change resulting in the key features of the current Cora dataset. A similar story also happens to the CiteSeer dataset.

In summary, the Cora, CiteSeer, and PubMed datasets are nowadays widely attributed to Sen et al. [14] as the original source. However, due to the complicated reasons listed above, the attribution of PubMed is wrong [54], while the attributions of Cora and CiteSeer have missed important earlier and

original sources [12, 13, 41]. The surprisingly chaotic citations of these very well-known datasets suggest that properly crediting the dataset contributors is not only a cultural or scholarship problem, but also a technical problem.

It is cumbersome for every researcher to go down the rabbit hole in the literature whenever they use a dataset. GLI instead attempts to provide a technical solution by having a dedicated bibliographic history section attached to the README file of each dataset. More importantly, this information is hosted on GitHub, which can be easily discussed and corrected, as what we have now may still miss important contributions to each dataset.

## C  Definitions of Graph Data Properties

Here we introduce the formal definitions of the graph data properties mentioned in Section 3.2. Given a graph $G = (V, E)$, where $V = \{1, 2, \ldots, N\}$ is the set of $N$ nodes and $E \subseteq V \times V$ is the set of edges. Denote $M = |E|$. Assume $X = \mathbb{R}^{N \times D}$ is the matrix of node features, where $D$ is the feature dimension. Also assume $Y = \{1, 2, \ldots, C\}^N$ is the vector of node labels, where $C$ is the number of classes.

### C.1  Basic

**Is Directed:** Whether the graph is a directed graph.

**Number of Nodes:** The number of nodes $N$.

**Number of Edges:** The number of edges $M$.

**Edge Density:** The edge density is defined as $\frac{2M}{N(N-1)}$ for undirected graph and $\frac{M}{N(N-1)}$ for directed graph.

**Average Degree:** The average degree is defined as $\frac{2M}{N}$ for undirected graph and $\frac{M}{N}$ for directed graph.

**Edge Reciprocity:** The edge reciprocity of a directed graph is defined as $\frac{\overleftrightarrow{M}}{M}$, where $\overleftrightarrow{M}$ denotes the number of edges pointing in both directions.

**Degree Assortativity:** The degree assortativity is defined as the average Pearson correlation coefficient of degree between all pairs of linked nodes.

### C.2  Distance

**Diameter:** The maximum pairwise shortest path distance in the graph.

**Pseudo Diameter:** The pseudo diameter approximates diameter, which serves as a lower bound of the exact value of diameter.

**Average Shortest Path Length:** The average of all the pairwise shortest path distance in the graph.

**Global Efficiency:** The efficiency between a pair of nodes is the multiplicative inverse of the shortest path distance and the global efficiency is the average efficiency of all pairs of nodes in the graph.

### C.3  Connectivity

**Relative Size of LCC:** The relative size of the largest connected component is defined as the ratio between the size of the largest connected component and $N$.

**Relative Size of LSCC:** The relative size of the largest strongly connected component is defined as the ratio between the size of the largest strongly connected component and $N$.

**Average Node Connectivity:** The local node connectivity for two non-adjacent nodes $u$ and $v$ is the minimum number of nodes that must be removed in order to disconnect them and the average node connectivity is the average local node connectivity of all pairs of two non-adjacent nodes in the graph.

## C.4 Clustering

**Average Clustering Coefficient:** The local clustering coefficient for node $u$ is defined as $\frac{2}{deg(u)(deg(u)-1)}T(u)$ for undirected graph, where $T(u)$ is the number of triangles passing through node $u$ and $deg(u)$ is the degree of node $u$; and defined as $\frac{2}{deg^{tot}(u)(deg^{tot}(u)-1)-2deg^{\leftrightarrow}(u)}T(u)$ for directed graph, where $T(u)$ is the number of directed triangles through node $u$, $deg^{tot}(u)$ is the sum of in degree and out degree of node $u$ and $deg^{\leftrightarrow}(u)$ is the reciprocal degree of $u$ and average clustering coefficient is the average local clustering of all the nodes in the graph.

**Transitivity:** The fraction of all possible triangles present in the graph, which is defined as $3\frac{\#triangles}{\#triads}$, where a $triad$ is a pair of two edges with a shared vertex.

**Degeneracy:** The least integer $k$ such that every induced subgraph of the graph contains a vertex with $k$ or fewer neighbors.

## C.5 Distribution

**Power Law Exponent:** The exponent parameter of a Power-law distribution that best fits the degree-sequence distribution of the graph.

**Pareto Exponent:** The exponent parameter of a Pareto distribution that best fits the degree-sequence distribution of the graph.

**Gini Coefficient of Degree:** The Gini coefficient of the the degree-sequence of the graph.

**Gini Coefficient of Coreness:** The Gini coefficient of the the coreness-sequence of the graph, where the coreness of a node $u$ indicates the largest integer $k$ of a $k$-core containing node $u$.

## C.6 Attribute

**Edge Homogeneity [20]:** The ratio of edges that connect nodes with the same node labels.

**Average Within-Class Feature Angular Similarity [20]:** Within-class angular feature similarity is $1 - angular\_distance(X_i, X_j)$ for an edge with its endpoints $i$ and $j$ with the same node labels and average within-class angular feature similarity is the average of all such edges in the graph.

**Average Between-Class Feature Angular Similarity [20]:** Between-class angular feature similarity is $1 - angular\_distance(X_i, X_j)$ for an edge with its endpoints $i$ and $j$ with different node labels and average between-class angular feature similarity is the average of all such edges in the graph.

**Feature Angular SNR [20]:** The ratio between average within-class feature angular similarity and average between-class feature angular similarity.

**Homophily Measure [22]:** The homophily measure is defined as

$$\hat{h} = \frac{1}{C-1}\sum_{k=1}^{C}[h_k - \frac{|C_k|}{N}]_+, \tag{1}$$

where $[a]_+ = max(a, 0)$, $|C_k|$ is the number of nodes with node label $k$ and $h_k$ is the class-wise homophily metric defined below,

$$h_k = \frac{\sum_{u:Y_u=k} d_u^{(Y_u)}}{\sum_{u:Y_u=k} d_u}, \tag{2}$$

where $d_u$ is the number of neighbors of node $u$ and $d_u^{(Y_u)}$ is the number of neighbors of node $u$ that have the same class label.

**Attribute Assortativity:** The attribute assortativity is defined as the average Pearson correlation coefficient of the attribute (class labels) between all pairs of linked nodes.

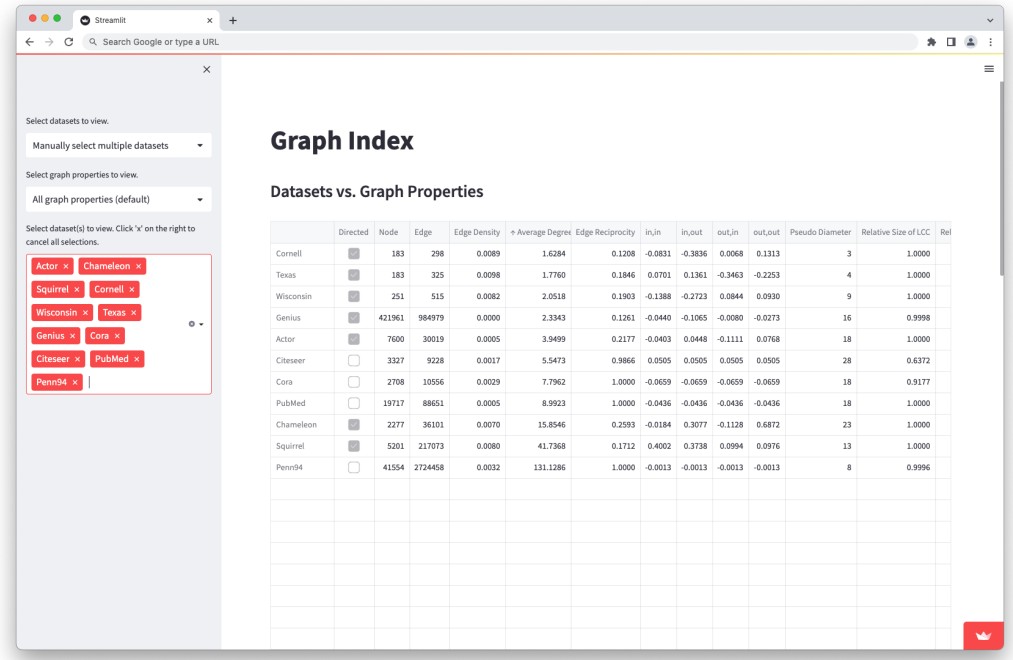

**Figure 4:** An interactive web page showing the graph properties for each dataset. The left sidebar shows which datasets are selected. The entries are sorted according to the average degree of nodes in this demonstration.

### C.7 Visualization

We create a web page to show the aforementioned graph data properties, as shown in Figure 4. We use Streamlit[21] to build and host the website. Users can select multiple datasets and graph properties, and sort by a graph property for a quick comparison.

## D Benchmark Experiment Setup

In this section, we describe more details of the experiment setup[22].

We set GCN [66], GAT [67], GraphSAGE [68], MoNet [69] , MLP, and MixHop [70] to have two layers in the benchmark setting. For LINKX [22], we set $MLP_A$, $MLP_X$ to be a one-layer network and $MLP_f$ to be a two-layers network, following Lim et al. [22].

In order to make a fair comparison, we adopt the same training configuration for all experiments. We use Adam [71] as optimizer for all models except LINKX. AdamW [72] is used with LINKX in order to stay the same with Lim et al. [22]. For all binary classification datasets (penn94, pokec, genius and twitch-gamers), we choose ROC AUC as evaluation metric. For other datasets, test accuracy is used.

Our implementaions of GCN, GAT, GraphSAGE and MoNet are based on DGL [10]. When implementing the models, we reserve default settings in DGL implementation as much as possible. For MixHop and LINKX, we adopt the implementation of Lim et al. [22]. The detailed settings for different models are listed below.

- GAT: Number of heads in multi-head attention = 8. leakyReLU angle of negative slope = 0.2. No residual is applied. Dropout rate on attention weight is the same as overall dropout.
- GraphSAGE: Aggregator type is GCN. No norm is applied.

---

[21]https://streamlit.io/cloud.
[22]Please see more details about how to use the benchmark code at https://github.com/Graph-Learning-Benchmarks/gli/blob/main/benchmarks/NodeClassification/README.md.

- MoNet: Number of kernels = 3. Dimension of pseudo-coordinte = 2. Aggregator type = sum.

- MixHop: List of powers of adjacency matrix = $[1, 2, 3]$. No norm is applied.

- LINKX: $MLP_A$, $MLP_X$ are both one-layer network and $MLP_f$ is a two-layers network. AdamW is used as optimizer. No inner activation.

**Hyperparameter tunning.**   Random search on the following hyperparameter tuning range is performed for every model.

- Hidden size: $\{32, 64\}$

- Learning rate: $\{.001, .005, .01, .1\}$

- Dropout rate: $\{.2, .4, .6, .8\}$

- Weight decay: $\{.0001, .001, .01, .1\}$

We generate 100 random configurations for each model, where each random configuration is run for 5 times on each dataset. The max training epoch number is 10000. We apply early stopping where training is stopped if the validation accuracy does not improve for 50 epochs. When training is finished, we load the weights of models with highest validation accuracy on the dataset. Test accuracy and standard deviation are reported in Table 1.

**Gradient Boosting Decision Tree (GBDT) models.**   We also include two GBDT models, Cat-Boost [73] and LightGBM [74], which are shown to be strong baselines [75–77]. We train both models for at most 1000 epochs with early stopping if validation accuracy does not improve for 100 epochs. We apply grid search on the following hyperparameters, and we have 5 independent trials for each hyperparameter configuration.

Hyperparameters for CatBoost:

- learning rate: $\{.01, .1\}$;

- depth: $\{4, 6\}$.

Hyperparameters for LightGBM:

- learning rate: $\{.01, .1\}$;

- number of leaves $\{15, 63\}$.

# E   Package Maintenance

This section outlines the designs of GLI that aim to ensure long-term viability and usability as an open-source project.

## E.1   Open Source License

GLI adopts the MIT License, aligning with our principle to favor broader application and trustworthiness of graph learning. By using MIT License, we only "require preservation of copyright and license notices. Licensed works, modifications, and larger works may be distributed under different terms and without source code."[23]

## E.2   Package Indexing

Currently, GLI provides a `setup.py` to facilitate the installation from the source. Moreover, we divide the package dependencies into three categories: `default`, `test`, and `doc` to meet different needs of users and potential contributors. We tested and successfully installed GLI on popular operating systems (Windows 11, MacOS with M1, Ubuntu, and CentOS). As a part of future work, we will use package indexing tools, including PyPI and Anaconda, to package the GLI project.

---

[23] https://chooselicense.com/licenses/mit/

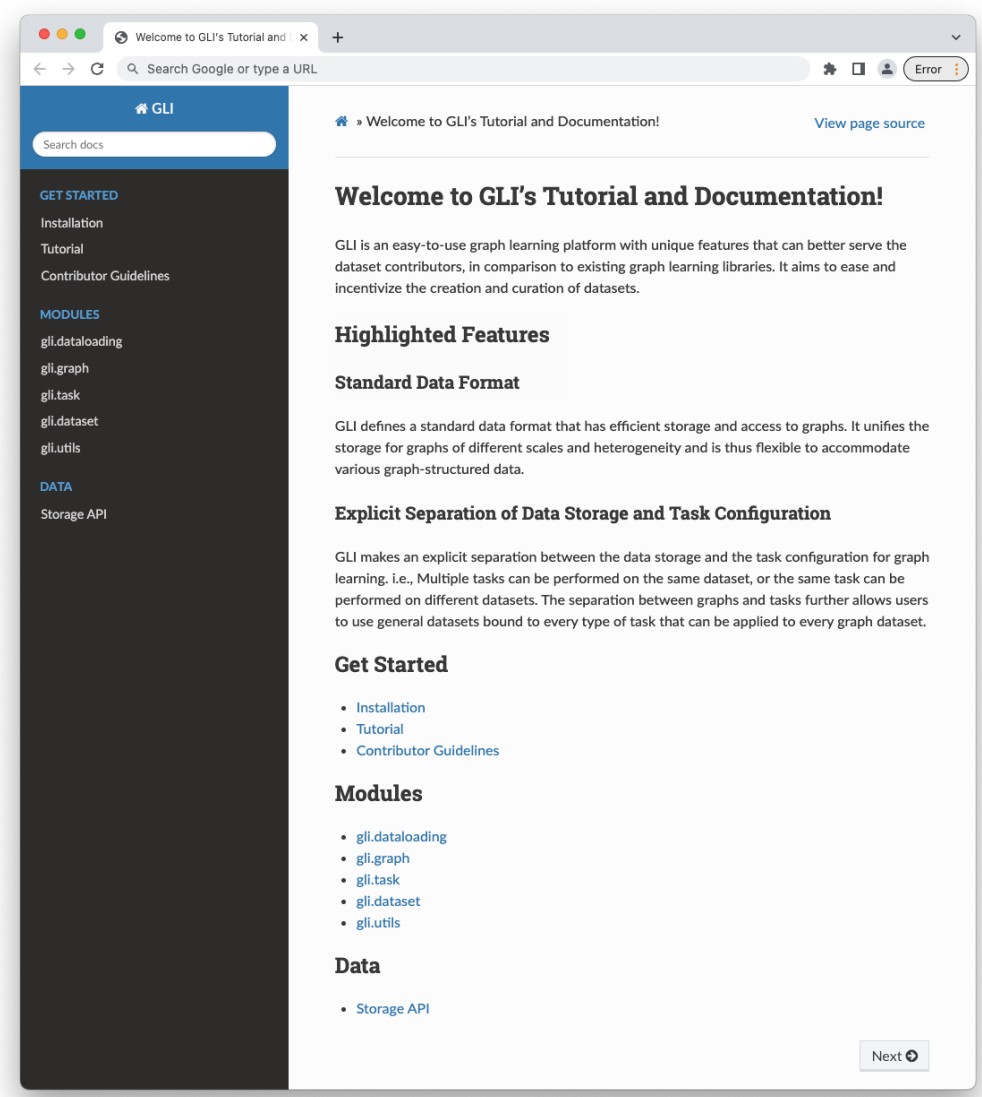

**Figure 5:** Preview of GLI document page.

### E.3  Documentation

**Automatic deployment.**    GLI uses sphinx[24] and autosummary[25] to generate API references automatically from docstrings in source codes. The docstrings are written in NumPy format[26] for better readability, in comparison to the common Sphinx format.

**Structure.**    Figure 5 shows the main page of GLI's documents. The web page has three main sections: "Get Started", "Modules", and "Data" as shown on the left toctree. The "Get Started" section contains an instruction on installation, and a tutorial for examples of basic usages and contributor guidelines. The "Modules" section contains the API references to core modules in GLI. The "Data" section illustrates GLI's file-based storage API.

---

[24]https://www.sphinx-doc.org/en/master/

[25]https://www.sphinx-doc.org/en/master/usage/extensions/autosummary.html

[26]https://numpydoc.readthedocs.io/en/latest/format.html

### E.4 Contributor Guidelines

We position contributor guidelines in two places: `CONTRIBUTING.md` in GitHub repository and "Contributor Guidelines" section in the aforementioned online document page. The contributor guidelines include the installation of the development environment, the steps to make contribution, and the development practices to follow. In particular, we distinguish three kinds of contribution: new dataset, new feature, and bug fix and ask contributors to follow different guidelines correspondingly.

### E.5 Tutorial

To flatten the learning curve for new users and potential contributors, we prepared a brief tutorial for their reference. The tutorial starts with an explanation of GLI's architecture, and follows with examples of data-loading and downstream tasks. For example, to train a GCN on Cora node classification task.

### E.6 Code Quality

GLI uses `pylint`, `pycodestyle`, and `pydocstyle` to ensure the code quality. Specifically, we have followed Google Python Style Guide[27] to configure the automatic linting and style checking tools. Moreover, they are enforced through both pre-commit hooks locally and continuous integration remotely. Besides, GLI uses `pytest` to help developers test whether a new patch violate the correctness of the codes. The testing is designed to cover all the core modules of GLI, including `gli.graph`, `gli.task`, `gli.dataset` and `gli.dataloading`. Users can run testing locally before they open a pull request. We also set up the continuous integration to run testing on GitHub, and enforce that a pull request must pass all tests before merging.

### E.7 Others

GLI uses `Makefile` to facilitate the development. Contributors can run `make test` to run the aforementioned tests locally to test the core modules on all datasets. Alternatively, one can specify a single dataset to test by `make pytest DATASET=<dataset name>`, which is a common scenario for dataset contribution.

---

[27] https://google.github.io/styleguide/pyguide.html

