# OpenReview forum: "Graph Learning Indexer: A Contributor-Friendly and Metadata-Rich Platform for Graph Learning Benchmarks"
_logconference.io/LOG/2022/Conference — LoG 2022 Oral_

### Official Review · Reviewer_Pakb · 2022-10-02

**Overall Score:** 6
**Confidence:** 4

**Review:**

Summary:
The authors propose a platform to index benchmark datasets for various graph learning tasks (e.g. node classification, link prediction). Special care is taken to incentivize contributors and to create a flexible, decoupled framework that can easily be extended for novel tasks.

Of note, the authors' core principle of properly assigning credit to each dataset creators/contributors is very commandable. We also appreciate the careful thoughts that went in decoupling datasets and tasks, making the addition of new tasks to a dataset easier. However, there is still some necessary additions to make GLI a valuable contribution to the community.

Comments:
- The term "metrics" for the "graph metrics" should be replaced as it has a very specific meaning in mathematics. We suggest using "graph statistics" or "graph properties" instead.
- While we appreciate the endeavour to make contribution as easy as possible, it feels that the resulting workflow is a bit tedious from a contributor stand-point. If we understand correctly, a contributor would first need to convert all data locally to the correct format using provided template script, then submit a PR containing all metadata to the github repository, and finally store the data itself to a separate repository not maintained by GLI. We wonder if this could perhaps be abstracted through a single submission interface? We also suggest it might be beneficial that GLI stores the data itself, hence ensuring its perennity and avoiding placing any responsibility on the contributor.
- The metadata of each dataset should include an associated LICENSE.
- It is unclear what requirements a dataset would need to meet for integration. We especially would like to see some ethic guidelines added to the repository. While this can sometimes be left to common sense, it is often useful to have clear public stance on these matters.
- While the text stresses how GLI incentivizes contributions as it is one of the key contribution of the authors, the article would benefit from the addition of a section on the data loading api of GLI. For instance, what format is the data when downloaded? How can it be interfaced with popular GML libraries such as PyG, DGL, etc.? Does GLI has some conversion modules to make integration seamless?
- We appreciate the addition of a wide range of graph statistics/properties, however without a web page, it makes it difficult to compare datasets at a glance. GLI would greatly benefit from such an addition whereby user can check included datasets and compare these according to one or more of the proposed statistics.
- (minor) "homophilious" is spelt "homophilous" and "non-homophilous" can be replaced by "heterophilous".
- (minor) It is a nice idea to have a suite of GML models benchmarked on new datasets. It is however unclear if Table 1 corresponds to the actual output of the functionality, or if it was prettified for this article. In any case, it might be more informative to include details on how to run the benchmarking in practice and what would be the actual output.
- (minor) in Table 2 (appendix), it would be useful to know at a glance what tasks is supported by each dataset.

---

### Official Review · Reviewer_SGpq · 2022-10-11

**Overall Score:** 8
**Confidence:** 4

**Review:**

**Overview/Summary:**

The paper presents Graph Learning Indexer (GLI), a benchmark curation platform for graph learning datasets and tasks. It claims to differentiate itself from existing benchmark curation efforts for graph learning such as OGB via two main objectives. Firstly, GLI aims to provide a contributor centric approach to collecting datasets enabled by lowering the difficulty of preprocessing datasets and tasks into a standardised format for submission as much as possible through guides and scripts. Second, GLI proposes a collection of meta-information consisting of elements such as graph characteristics, target characteristics, and benchmarks by standard models to construct a knowledge base of the datasets+tasks. I would also highlight the flexible data API which allows for a more flexible dataset + task setup. The paper is dedicated to explaining this contribution pipeline, the file based storage API that is more flexible than existing code-based solutions in principle, and the collection of meta-information and what is done with it.

**Review start:**

My review of the contributions in this paper is based on the premise that a resource contribution such as an open source software library (and benchmark curation platform in this case) should not be assessed in the same manner as a research paper. This is primarily as the judgement of novelty is different between a resource contribution and a research contribution where we typically look for novel findings, generalisations and methodologies or interesting observations. In contrast, this review will attempt to assess the software library on its potential impact on the GRL research community and its adherence to good software engineering practices. Namely, the package will also be assessed on aspects such as its documentation, its contributor and community guidelines, software engineering practices, how easy it is to navigate and understand the codebase, and more alongside the novelty, breadth and significance of the contribution.

In my assessment of GLI I planned to:
- Read the paper
- Attempt to install the package
- Attempt to find documentation
- Run through any basic tutorials or quickstarts
- Attempt to run tests and look at the testing approach
- Look for contributors guidelines
- Look for evidence of adhering to coding standards
- Checked degree of openness (Licences etc.)

Unfortunately as the anonymous_github platform currently does not allow the cloning of the repository (as described in https://github.com/tdurieux/anonymous_github/issues/24) I was unable to perform steps 2, 4, 5. However as this is not the author(s)’s fault I read through the codebase looking for evidence of good practice on those points.

*Paper-specific pros/cons*

Pros:

- Proposes a scalable approach to soliciting datasets from the bottom up, with effort made to lower the difficulty of submitting datasets to the platform via a unifying data API with guides+scripts to transform potential datasets to them.
- Good incentives for the dataset contributors, especially in regard to crediting the original dataset source in addition to the papers that popularised them.
- Well explained pipeline for preparing, processing, and submitting datasets for indexing. Particularly useful is the testing and code quality check ensuring correctness of the dataset format as explained in section 2.1.2
- Proposes a useful abstraction for datasets and the associated learning tasks on it. It is described well in the paper and Figure 1 is particularly effective for illustrating the data management API.
- Example in paper highlighting ease of use.
- Good appendix with useful definitions for the meta-datas, and benchmark experimental setup.

Suggestions/Cons:

- Not clear what license is attributed to the contributed datasets (this would be good to have in the main text).
- This is more suggestion rather than a con, but as an open source resource paper it would be beneficial to have a section on how the long term viability of the project is ensured. (see below text for more detail)
- Again more suggestion than con, it would be good to have brief statements about the code of conduct / ethics notes about the datasets, and how these may be handled.

*Software specific pros/cons:*

Pros:

- Code quality guidelines and evidence of tools for this. The authors may enhance this further by adding pre-commit hooks for contributors.
- Contributors guidelines given
- MIT License at the root of the project
- Tests are present. Contributors notes contains information on using the testing framework.
- Already has a good starting collection of common datasets

Cons:

- Currently no binaries are released on standard package indexes (which may be for double blind purpose, so please ignore if this is the case). Installation is done from source. This is ok, but packaging may ease installation for a broad range of users especially with regard to dependencies as time progresses. It also helps separate stable versions with development versions of GLI.
- Documentation not found in submitted repository. It would be nice to have an accompanying website with API reference (and be able to build it locally). It would be useful to have a series of longer annotated tutorials on such a page. This can be done for free using the a sphinx+autodocs documentation setup combined with the ReadTheDocs hosting service which is standard approach for many OSS ML packages (e.g. PyG, PyG Temporal, Geo2DR, SquidPy to name a few).  The benefits of this combination is that API reference pages would be created automatically from the docstrings present in the code.
- Information can be found on the ReadTheDocs website: https://readthedocs.org/
- Sphinx + AutoDocs: https://www.sphinx-doc.org/en/master/usage/extensions/autodoc.html
- (Related to above) I feel that there should be more docstrings throughout the code, however I appreciate that a static analysis tools (pydocstyle) has been used to check the formatting.
- A separate code of conduct in the repository would be good to promote healthy community and guidelines for handling abuse. Github encourages these and has a guide here: https://docs.github.com/en/communities/setting-up-your-project-for-healthy-contributions/adding-a-code-of-conduct-to-your-project

*Questions:*

- Why do some datasets have associated notebooks (eg. ogbg-molsider) and others not?
- I see that most of the actual data of the datasets will be stored on google drives, which is a sensible choice for robustness and availability, compared to hosting done by the contributors. Is there any policy about the governance/rights dataset contributors will have over their data once submitted? This leads to the next question.
- What kind of licenses will contributed datasets be licensed under?
- Are datasets versioned for updates? Or do they become new datasets/entries in the indexer?
- Will there be meta data concerning ethical notes that can be made retrospectively to deprecate datasets?. For example, a dataset may be deprecated on ethical grounds such as the Boston dataset that used to be part of many packages such as SkLearn (https://scikit-learn.org/stable/modules/generated/sklearn.datasets.load_boston.html) that is now rightfully removed on ethical grounds.

Below are just a few more additional comments complementing the bullet points above, and the decision at the bottom.

Overall, the paper does a good job of explaining its contributions and highlighting its differences to existing benchmark dataset curation efforts. Particularly notable from a technical standpoint is the data API that decouples the dataset from the learning task allowing for more flexible and storage efficient approach to storing datasets. I believe this will have a positive effect on end-users in forcing acknowledgement and specification of the exact task they wish address with their ML solutions on the dataset. From a research community standpoint the package attempts to address the rampant misattribution and citations given to papers that popularise datasets whilst the original contributors are largely forgotten. The amount of text and effort dedicated to addressing this point is appreciated.

One thing I believe may benefit the paper is the addition of a “Package maintenance” section, which could be in the main text or in an appendix. This section would outline the steps taken by the authors to ensure the long term viability and usability of the software package. Such a section would describe the modern software engineering practices used to promote (and in some sense promise through the paper) the long term robustness and usability of the project for both users and contributors. This would contain notes about the presence of the open source licence used, package indexing, and contributors guidelines. It would have notes on documentation (once completed): how the API reference is built, what practices are expected on the documentation of new code additions, etc. Another point would be the code quality practices followed, whether automatic linting tools are used, and whether that is enforced by a pre-commit hook. Finally, it may also contain statements about how the code is tested, the coverage goals, whether continuous integration is setup for contributors and release pipelines. This may sound tedious to write up, but also goes a long way to showcase the effort and care taken to make the package robust, reliable, and build trust in the code (and maintainers).

Based on the bullet points and general comments above I recommend a weak acceptance. The positives of the current state of the paper and software as well as the potential positive impact of the proposed GLI platform on the GRL community outweigh many of the negative points. However, I hope the authors take on my comments and questions made on packaging, documentation, and licensing (of the contributed datasets) to really make the project flourish.

Misc notes:

- Line 180: “... identify potential biases and trustworthy issues existed in the datasets…” maybe better rephrased as “... identify potential biases and issues regarding the trustworthiness in the datasets…”

---

### Official Review · Reviewer_6eeH · 2022-10-17

**Overall Score:** 6
**Confidence:** 4

**Review:**

## Summary
This paper proposes GLI, a contributor-centric benchmark curation platform for graph representation learning. In particular, GLI uses file-based data management API design such that (1) dataset contributors can easily convert their original dataset format into GLI format, (2) a unified data loading and task definition can be performed, and (3) data characteristics including graph average degree etc. can be calculated without the effort from dataset contributors. Beyond this, GLI support a benchmark indexing system to provide additional informations about various datasets, which are informative for dataset users when they select the suitable dataset.

## Reasons for score
Overall, I vote for weak acceptance. 1. The idea of building up a contributor-centric benchmark curation platform is interesting and beneficial to the graph learning community and can save a lot of maintaining efforts from dataset contributors. 2. Additionally, to make sure the uploaded datasets are clean and meaningful, the authors claim to provide guidances and peer reviews for the uploaded datasets, and also provide indexing informations for different datasets. 3. The writing of this paper is clear and easy to follow.

## Pros:
As I mentioned above in reasons for score.

## Cons and Questions:
1. I noticed that when dataset contributors upload their datasets, they need to write some code to convert the data format into GLI format through .py or inpynb file. I tend to believe this is a not-so-hard task, but is there a way to further eliminate efforts from contributors?

2. As you mentioned in the paper, uploaded datasets will be peer-reviewed. This is a huge burden for organizers but is essential for the quality of uploaded datasets.

3. Quality control of newly uploaded datasets is my main concern.

---

### Official Review · Reviewer_8mfE · 2022-10-18

**Overall Score:** 8
**Confidence:** 5

**Review:**

The paper proposes a new graph management system that complements existing databases in several aspects: separation of data and tasks, easy API, documentation, automated tests, version control, metrics, etc. -- which facilitates more friendly and interactive experience for the graph contributors. Given growing interest in developing new models for graph structured data it's of paramount value to have a curated database that should further facilitate graph learning research.

Strong points:
* The paper is well-written and explains its contribution in a concise and clear manner;
* The paper takes into account strong and weak points of existing databases and proposes solutions to address the problems;
* The paper implements a benchmark comparison of existing graph methods;
* The paper includes a range of graph metrics which can be computed for a given dataset.

Weaknesses:
* The paper does not store the proposed datasets but instead relies these datasets are stored on the third-party server. It risks to lose the dataset in the future if third-party server removes the dataset.
* The list of graph methods in Table 1 is far from being comprehensive with many novel SOTA methods existing. Perhaps, it requires a service that would allow any new model to be tested automatically on your datasets.
* Similar to PapersWithCode or OGB, it's valuable to have a leaderboard of methods for each dataset.

Considering that there is increasing interest in graph structured data in machine learning community, a new curated database of graphs is instrumental for further acceleration of research in this area, therefore I recommend to **accept** this paper. A humanly-curated database of graphs can be of value not only for the academic research but could also act as a standalone commercial service that delivers high quality data to the industrial applications.

Questions:
1. Addressing the weak point 1 have you considered storing all data in a dedicated server, either locally or in the cloud?
2. Right now you include about 40 datasets which is not a lot for all graph applications. How do you plan to collect more datasets in a proactive manner? How many datasets do you believe is sufficient for all graph applications?
3. Why in appendix A you don't consider node/graph regression problems?

Suggestions:
1. In the list of baselines I would necessarily put GBDT solutions. It's an often overlooked baseline for graph methods, however, it often performs better than any of the neural network solutions. See [1,2,3].
2. I would suggest to go over recent papers to collect the datasets from there. Examples of datasets not included in the GLI: [1, 4].


[1] Boost then Convolve: Gradient Boosting Meets Graph Neural Networks https://arxiv.org/abs/2101.08543
[2] Does your graph need a confidence boost? Convergent boosted smoothing on graphs with tabular node features https://arxiv.org/abs/2110.13413
[3] High Performance of Gradient Boosting in Binding Affinity Prediction https://arxiv.org/abs/2205.07023
[4] New Benchmarks for Learning on Non-Homophilous Graphs https://arxiv.org/abs/2104.01404

---

### Meta-Review · Area_Chair_7v1N · 2022-11-17

**Confidence:** 5
**Recommendation:** Accept for spotlight

**Meta Review:**

This paper proposes a platform for curating graph learning datasets.  Reviews for this paper are (perhaps curiously) overwhelmingly positive.

Strengths:
+ S1. Reviewers liked the idea of a 'friendlier OGB' that allowed more easy dataset curation.
+ S2.  Reviewers like the emphasis on properly assigning credit to dataset creators.
+ S3.  The design of the API allows abstraction between tasks, and the data itself.

Weaknesses:
- W1.  Reviewers found weak points in the actual project itself (props to them for inspecting it).  However authors had good responses, and addressed a lot of the issues.

In summary, there's basically no negative points against this paper and it's a clear candidate for acceptance.

---

### Decision · Program_Chairs · 2022-11-23

Accept (Oral)